# Taming the chaos gently: a predictive alignment learning rule in recurrent neural networks

Toshitake Asabuki[1,2] ✉ & Claudia Clopath [3] ✉

Recurrent neural circuits often face inherent complexities in learning and generating their desired outputs, especially when they initially exhibit chaotic spontaneous activity. While the celebrated FORCE learning rule can train chaotic recurrent networks to produce coherent patterns by suppressing chaos, it requires non-local plasticity rules and quick plasticity, raising the question of how synapses adapt on local, biologically plausible timescales to handle potential chaotic dynamics. We propose a novel framework called "predictive alignment", which tames the chaotic recurrent dynamics to generate a variety of patterned activities via a biologically plausible plasticity rule. Unlike most recurrent learning rules, predictive alignment does not aim to directly minimize output error to train recurrent connections, but rather it tries to efficiently suppress chaos by aligning recurrent prediction with chaotic activity. We show that the proposed learning rule can perform supervised learning of multiple target signals, including complex low-dimensional attractors, delay matching tasks that require short-term temporal memory, and finally even dynamic movie clips with high-dimensional pixels. Our findings shed light on how predictions in recurrent circuits can support learning.

Humans and animals exhibit remarkable capabilities in learning and generating complex behaviors essential for a wide range of tasks. Indeed, the brain can accurately acquire and recall complex sequences, from the order of words in a sentence to highly skilled motor behaviors. The brain's capacity to flexibly process sequential data and generate complex outputs is underpinned by the continuous, coordinated activity of neural networks, which integrate temporal and spatial information to precisely control its outputs.

Recurrent Neural Networks (RNNs) are powerful computational models that can capture and process sequential information while exhibiting complex dynamics[1-4]. Cortical circuits often exhibit chaotic spontaneous activity[5], and RNNs can generate such dynamics through feedback loops[6-11]. While this chaotic behavior provides a variety of basis functions for network dynamics, the question arises as to how it can be transformed into desired patterned dynamics. The FORCE learning, which originated from reservoir computation[12-20], has been

widely used as an algorithm for learning networks that exhibit chaotic behavior[21]. However, this approach relies on non-local weight updates with fast synaptic changes, seemingly lacking biological plausibility. How biologically plausible learning algorithms for recurrent neural networks can effectively leverage their rich dynamical properties to efficiently learn and recall complex sequential information remains an elusive challenge[21-24], especially when the networks show chaotic dynamics.

In this paper, we introduce "predictive alignment", an alternative learning framework designed to train recurrent neural networks over a variety of complex tasks while overcoming the limitations of existing methods. Our proposed learning rule modifies plastic recurrent connections to predict output feedback signals, while aligning these predictive dynamics with existing chaotic spontaneous dynamics (arising from fixed recurrent connections), which in turn suppress the chaos efficiently and improving network performance. The key innovation of

[1]RIKEN Center for Brain Science, RIKEN ECL Research Unit, Wako, Japan. [2]RIKEN Pioneering Research Institute, Wako, Japan. [3]Department of Bioengineering, Imperial College London, London, UK. ✉e-mail: toshitake.asabuki@riken.jp; c.clopath@imperial.ac.uk

predictive alignment lies in its ability to perform online and local supervised learning through prediction, enabling the network to learn multiple target signals efficiently and robustly. We demonstrate that predictive alignment can successfully train networks to generate diverse complex target signals with nonlinear dynamics, such as the chaotic Lorenz attractor, delay-matching tasks that require short term memory of temporal information, and high-dimensional spatio-temporal patterns in a movie clip. The proposed learning rule not only sheds light on how predications can guide learning in circuits, but offers a biologically plausible solution for training powerful recurrent networks in various applications.

## Results

### Recurrent neural network

We first considered a rate-based recurrent network (Fig. 1, see spiking network in Supplementary Figs. 10 and 11). The network obeys the following dynamics:

$$\tau \frac{d\mathbf{x}}{dt} = -\mathbf{x} + \mathbf{J}\mathbf{r} \qquad (1)$$

where $x$ is a vector of membrane potentials of network units and $\tau$ is a time constant. The variable $\mathbf{r}$ is a firing rate vector, defined as $\mathbf{r} = \tanh(\mathbf{x})$. We will consider network dynamics with external drives and noise. The matrix $\mathbf{J}$ is a recurrent connectivity and is assumed to be a summation of two types of matrixes, $\mathbf{M}$ and $\mathbf{G}$:

$$\mathbf{J} = \mathbf{M} + \mathbf{G} \qquad (2)$$

Although both types of connections are assumed to be initially generated with Gaussian distributions, they have some division of labor. First, we assumed that $\mathbf{M}$ is a plastic but initially weak connection, while $\mathbf{G}$ is a strong and fixed connection of which the large elements lead to chaotic network activity prior to learning[21,25] (Methods). The recurrent plasticity rule we propose in this paper is applied to $\mathbf{M}$, while $\mathbf{G}$ is always static. Second, these connections are assumed to have different sparseness: $\mathbf{M}$ has full connections while $\mathbf{G}$ has a sparse connection (Methods). After learning, $\mathbf{M}$ will suppress the chaotic spontaneous activity.

The output of network, or a "readout" is assumed to be a weighted linear summation of network activities:

$$z = \mathbf{w}^T \mathbf{r} \qquad (3)$$

where $w$ is a readout weight vector and $T$ is a transpose. Multiple readouts can be defined, each with its own set of weight vectors.

### The predictive alignment learning rule

Similar to standard gradient descent rules, readout weights were trained to minimize the error between the target signals $f$ and the model's output $z$: $\mathcal{L}_{\text{out}} = \frac{1}{T} \int \| f(t) - z(t) \|^2 dt$, where $T$ is a duration of learning phase. As we are considering online learning rule, at each time step, the resulting weight update rule can be written as:

$$\Delta\mathbf{w} = \eta_W [f - z]\mathbf{r} \qquad (4)$$

where $\eta_W$ is a learning rate. For simplicity, the time dependence notation has been removed. The above rule for the readout weights is a standard least-mean-square (LMS) or also called the Delta-rule, hence not novel.

Predictive alignment is a new learning rule for the recurrent weights. Unlike most learning rules for recurrent networks, which minimize the same cost function for the outputs (i.e., $\mathcal{L}_{\text{out}}$ defined above), the aim of the predictive alignment is to predict a feedback signal from the readout unit with the recurrent dynamics, while aligning predictive recurrent dynamics to the chaotic dynamics. More precisely, the plastic recurrent connectivity $\mathbf{M}$ is asked to minimize the cost function shown below:

$$\mathcal{L}_{\text{rec}} = \frac{1}{2T} \int dt \|\mathbf{Q}z - \mathbf{M}\mathbf{r}\|^2 - \frac{\alpha}{T} \langle \mathbf{G}\mathbf{r}, \mathbf{M}\mathbf{r} \rangle \qquad (5)$$

Here, the first term of the cost function is a deviation between the recurrent dynamics $\mathbf{M}\mathbf{r}$ and the feedback signal $\mathbf{Q}z$, where $\mathbf{Q}$ is a static feedback connection. It should be noted that feedback from readout appears only in the learning rule, hence does not affect network dynamics directly. Minimizing this first term in the cost function requires the recurrent dynamics to predict the feedback signal. The second term is a regularization term, which plays a crucial role to suppress the chaos efficiently by aligning the predictive recurrent dynamics $\mathbf{M}\mathbf{r}$ to the chaotic dynamics $\mathbf{G}\mathbf{r}$. We will show the effect of regularization in detail later. Unless other specified, we will assume $\alpha = 1$ throughout the paper. The resulting online learning rule which minimizes the cost function $\mathcal{L}_{\text{rec}}$ can then be written as:

$$\Delta\mathbf{M} = \eta_M \left[ \mathbf{Q}z - \hat{\mathbf{j}}\mathbf{r} \right] \mathbf{r}^T \qquad (6)$$

where we defined a regularized recurrent prediction as $\hat{\mathbf{j}}\mathbf{r}$ with $\hat{\mathbf{J}} = \mathbf{M} - \alpha\mathbf{G}$ (Methods).

In summary, we have proposed a set of plasticity rules for training chaotic recurrent neural networks. The readout weights are simply modified to minimize the error between the actual output and the target signal, while the recurrent connections are modified to minimize the feedback prediction error while aligning

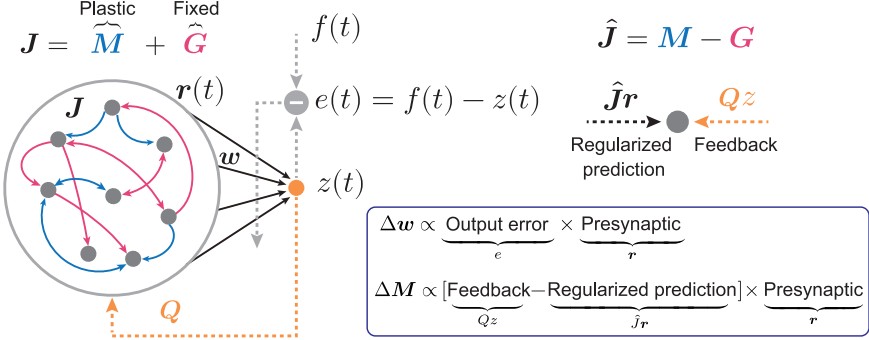

**Fig. 1 | Recurrent neural network and predictive alignment rule.** The network consists of recurrent layer and a readout unit. Multiple readouts will be considered later, yet only a single readout is illustrated in this example. Readout weight vector **w** is trained to minimize the error between a target and the readout activity (left figure, gray dashed arrows). Recurrent connection **J** is a summation of plastic yet initially weak connections **M**, and strong and fixed connectivity **G**. The plastic recurrent connections **M** is trained to minimize the error between feedback from output through random weights **Q** and a recurrent prediction **Mr**, while aligning predictive recurrent dynamics to the chaotic dynamics.

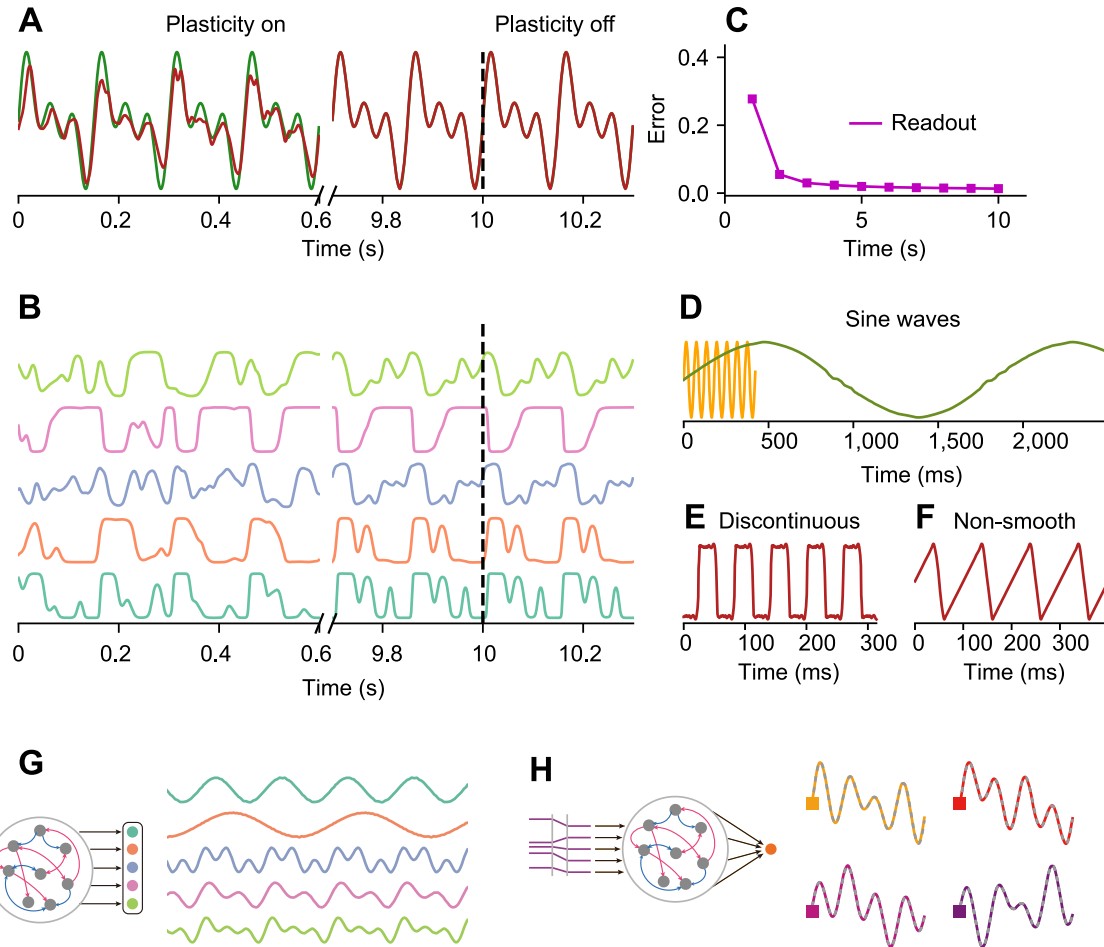

**Fig. 2 | Examples of predictive alignment. A** During the early phase of learning, the model output (red) deviated from the desired target (green), while it matched the output after sufficient training even after the plasticity was turned off. **B** The network activities initially showed chaotic activities and then was transformed to coherent activities after training. **C** Root mean squared error between the target and the output during training. **D**–**F** Examples of output activities trained to generate various simple target patterns. **D** The network learned sine waves $y = \frac{3}{2}\sin\left(\frac{2\pi t}{T}\right)$ with different frequencies ($T = 6\tau$; orange and $T = 200\tau$; green). **E** The model learned discontinuous step-like target. **F** The model learned non-smooth sawtooth pattern. **G** A network with five readout units was considered (left). Such network could generate five distinct target patterns simultaneously (right). **H** A network receiving external control inputs (left) generates corresponding output patterns (right; colored lines). The corresponding target signals are shown as gray dashed lines superimposed on the targets.

the predictive recurrent and chaotic dynamics—hence the name predictive alignment.

## Toy examples of learning with the predictive alignment rule

We first show some simple toy examples to demonstrate that predictive alignment can learn various forms of desired target signals. For the sake of simplicity, we assume periodic target signals in the toy examples, but we will consider non-periodic targets later. During the early phase of learning (Fig. 2A; 0-0.6 s), the readout activity (Fig. 2A; red trace) showed a mismatch from the target signal (Fig. 2A; green trace), indicating that the readout activity did not follow the target signal at this early stage. This is in contrast to the behavior observed in the FORCE learning paradigm, which consistently clamps the output to the target through recursive least squares (RLS). As learning progressed, the readout error decreased monotonically and the readout activity became similar to the target function (Fig. 2C). Once sufficient learning had occurred, the readout activity showed dynamics that were well matched to the target signal, even when plasticity was turned off (Fig. 2A; right side of vertical dotted line). Furthermore, while the recurrent units initially showed chaotic behavior, the activities became coherent and structured after sufficient learning (Fig. 2B).

Next, we examined how learning in the network changes the structure of the eigenvalue spectrum of the recurrent weight matrix. Before learning, the eigenvalues were uniformly distributed in a circle in the complex plane because the initial connectivity was a random matrix (Supplementary Fig. 1; left). After learning, while the connections had most of their eigenvalues within a circle in the complex plane, some pairs of leading eigenvalues with large real parts appeared (Supplementary Fig. 1; right). Such outliers in the spectrum generated by low-dimensional perturbations change the dynamics[26,27].

We further show that predictive alignment can learn more complex and diverse target patterns. The network can learn sinusoids with small (Fig. 2D orange) or much larger (Fig. 2D green) periods, as well as discontinuous (Fig. 2E) and even non-smooth (Fig. 2F) target signals.

In the above examples, we have demonstrated the performance of the network using simple examples. In particular, the network architecture considered above had only a single readout unit and learns only a single target signal. To see whether predictive alignment still works for multiple targets, we first extended the learning paradigm beyond a single readout unit and make the predictive alignment face complex scenarios involving multiple readout units and corresponding target signals. Here, each recurrent unit's plasticity was modulated by a weighted combination of all output signals (Methods). In our

simulation, we considered five periodic target signals, each was presented to one of readout units as a corresponding target signal. We found that the trained model exhibits five output patterns, distinctly aligned with the specified targets. This result shows the network's ability to learn and generate diverse output patterns, without requiring that feedback signals are constrained to subpopulation-specific interactions; in our simulation, they indeed were all-to-all interactions across the entire recurrent population.

Having demonstrated that a network could learn multiple target signals, we wondered whether the network could perform input-output transformation tasks. To test this, we introduced four static control input patterns to the network units (Fig. 2H). Each desired output function was paired with a corresponding input pattern. The inputs were randomly assigned constant values, without temporal information. They acted only as switches to generate specific output functions. We found that such a network with a single readout unit could learn to generate different outputs depending on the given control input patterns (Fig. 2H).

Next, to confirm the robustness of the model, we fed the network with different strengths of noise while training the patterned target signal. We confirmed that while the model's output error increased as the noise intensity was increased, the degree of increase was sufficiently small compared to FORCE (Supplementary Fig. 2).

We further investigated the robustness of the model with respect to its hyperparameters. The results were robust against variations of the learning rate (Supplementary Fig. 3A), the initial strength of the plastic recurrent connections, (denoted by the scale "g", see Methods, Supplementary Fig. 3B) and the connection probability of M (Supplementary Fig. 3C).

In summary, we have tested our predictive alignment learning rule over various forms of target signals and multiple readouts. Further, the model can transform static input signals to the transient output patterns by modifying the recurrent connectivity.

## Recurrent prediction aligns to the chaotic dynamics

To understand the mechanism underlying the predictive alignment, we next analyzed the effect of recurrent plasticity in the network undergoing learning of a simple periodic signal (as shown in Fig. 2A). Learning of the recurrent connections minimizes of error between the regularized recurrent dynamics and the feedback signal (Eq. 6). The error therefore dropped monotonically and a plastic recurrent dynamics ($\hat{\mathbf{j}}\mathbf{r}$) gradually matched to the feedback signal ($\mathbf{Q}z$) (Fig. 3A, B). We would emphasize that the term $\hat{\mathbf{j}}\mathbf{r}$ is the regularized recurrent prediction (see Eq. 6). Recall that minimization of our cost function (Eq. 5) requires alignment of the predictive recurrent dynamics $\mathbf{M}\mathbf{r}$ to the chaotic dynamics $\mathbf{G}\mathbf{r}$. In the following, we will call such an alignment of two types of dynamics "recurrent alignment".

We next demonstrate the role of recurrent alignment in our plasticity rule. To this end, we first calculated the output error over various degree of regularization parameter $\alpha$. We found that increasing the value of $\alpha$ resulted in a reduction in the model output error relative to the target function (Fig. 3C). Interestingly, the variance of the error over multiple simulations also decreased with higher values of $\alpha$. These results suggest that larger $\alpha$ increases the accuracy and stability of learning. To further understand the mechanism underlying these results, we first calculated the correlation term in the cost function (i.e., $\langle \mathbf{G}\mathbf{r}, \mathbf{M}\mathbf{r} \rangle$) over the entire learning period in two cases. For the first case, we trained the network with $\alpha = 1$, indicating that the learning rule reduces the error under a trade-off with the regularization term (aligned case). In contrast, in the second case, the recurrent connections were trained with $\alpha = 0$, hence such trade-off was not considered (control case). As expected, the value of correlation grew in

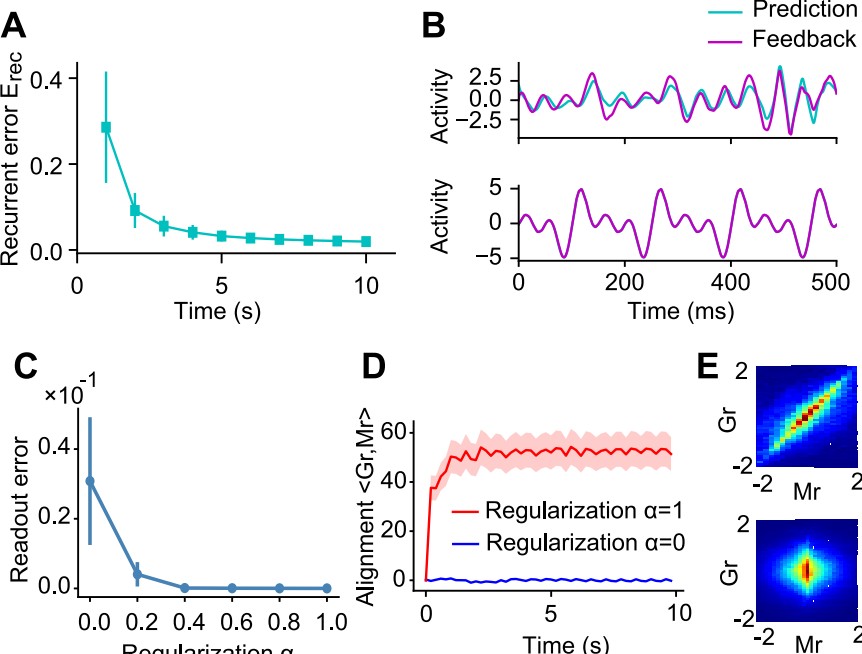

**Fig. 3 | Network mechanism of predictive alignment. A** The recurrent error

$$E_{\text{rec}} := \sqrt{\int dt \sum_i \left[ (\hat{\mathbf{j}}\mathbf{r})_i(t) - (\mathbf{Q}z)_i(t) \right]^2 / (NT_{\text{seg}})}$$

between the regularized recurrent prediction $\hat{\mathbf{j}}\mathbf{r}$ and the feedback signal $\mathbf{Q}z$ during training. Here, $N$ is the number of units in the network and $T_{\text{seg}}$ denotes the duration of each time segment, with the entire time period equally divided into 10 segments. The time integration was performed over the duration of each segment. **B** Example dynamics of the regularized recurrent prediction $\hat{\mathbf{j}}\mathbf{r}$ and the feedback signal $\mathbf{Q}z$ during early (top) and late (bottom) phases of learning are shown. **C** Root mean squared errors between the target and the trained output over different values of the regularization parameter are shown. **D** The dynamics of the correlation between the plastic recurrent $\mathbf{M}\mathbf{r}$ and the chaotic $\mathbf{G}\mathbf{r}$ dynamics are shown for with (red) and without (blue) regularization. **E** Joint distributions of plastic recurrent $\mathbf{M}\mathbf{r}$ and chaotic $\mathbf{G}\mathbf{r}$ dynamics are shown with (top) and without (bottom) regularization. In (**A**, **C**) error bars stand for s.d.s over 20 independent simulations. In D, shaded areas represent s.d.s over 10 independent simulations.

the aligned case, while such behavior was not observed in the control case (Fig. 3D, E). Furthermore, we found that in the aligned case, the network's Lyapunov exponent was shifted further toward the negative side compared to the control case, indicating more effective suppression of chaos (Supplementary Fig. 4).

Altogether, these results indicate that the predictive alignment suppresses the chaotic activity more efficiently by aligning the recurrent prediction to the chaotic dynamics, allowing for robust computation.

## Learning performance around the edge of chaos

While we have introduced a static strong recurrent connectivity **G**, the role of spontaneous chaotic activity generated by such strong connectivity remains to be explored. Following the standard rate-based recurrent network, the strength of static connectivity within the network was scaled by a factor g[25]. Recurrent networks with $g < 1$ produce decaying activity in the spontaneous activity regime, while $g > 1$ leads to irregular chaotic spontaneous activity[25]. In FORCE learning, initial networks with g just above such a critical point, the so-called "edge of chaos", show the best performance for learning.

We wondered to what extent the performance of our model depends on the strength of the scaling factor g. To test this, we first measured the average root mean square (RMS) error after training, across different networks with different values of $g$. Similar to FORCE, the error between the target function and the output of the network shows best performance when the scaling factor is just above 1.0 (i.e., the edge of chaos) (Supplementary Fig. 5A). We found that the RMS error between recurrent prediction and the feedback also showed a minimum value when the scaling factor $g$ was around the edge of chaos (Supplementary Fig. 5B). The model performs best at the edge of chaos because it facilitates the richness of network dynamics, which in turn provides diverse basis functions from which the output units can readout the appropriate dynamics.

We then measured the strength of the output weight vector and the recurrent connection matrix. Large values of the weights can lead to training instability and also make the network more sensitive to noise, indicating that the network is losing robustness (Supplementary Fig. 5C, D). We found that both weights had a minimum when the network is on the edge of chaos initially.

Taken together, these results suggest that predictive alignment produces accurate and robust output when the initial network state is on the edge of chaos. This chaos is then tamed gradually by the recurrent alignment (see Fig. 3).

## Structured diversity at the edge of chaos enhances learning

We further asked what is the role of edge of chaos. Optimal learning is thought to require a balance between representational diversity and dimensionality. High diversity allows the network to encode rich information, while low dimensionality ensures compact representations. To quantify this trade-off, we measured the entropy of the eigenvalues of the correlation matrix ($H_\lambda$) as an indicator of representational diversity, and the square root of the participation ratio ($\sqrt{PR}$) as a measure of the network's effective dimensionality (see Methods). We computed the ratio of these two measures,

$$\text{Efficiency} = \frac{H_\lambda}{\sqrt{PR}} \qquad (7)$$

and found that this ratio was maximized around the edge of chaos. In the subcritical regime, the participation ratio was low, indicating that activity was concentrated in a low-dimensional subspace, reducing the network's ability to learn diverse representations (Supplementary Fig. 6B). In contrast, in the chaotic regime, the participation ratio was high, but the eigenvalue entropy decreased, suggesting excessive dispersion of information, leading to loss of meaningful

structure (Supplementary Fig. 6A). At the edge of chaos, both measures were balanced, enabling the network to achieve the highest learning performance (Supplementary Fig. 6C). These findings suggest that learning is most efficient when the network exhibits rich yet structured dynamics, balancing representational diversity and stability.

## Fixed-point attractor analysis in the network

To gain further insight into how the proposed network reorganizes chaotic activity into the desired dynamics, we examine a network consisting of three readout units, each corresponding to the state of an independent memory bit[28]. The state of each output is determined by transient pulses from the corresponding input units. At random intervals, each input unit generates a transient pulse that can be either $+1$ or $-1$. These pulses affect the corresponding output unit, causing it to switch or maintain a value of either $+1$ or $-1$. Once the output value is set, it remains fixed until the arrival of the next pulse from the input unit. We trained a network ($N = 500$) to perform the task using the predictive alignment rule. We found that the trained network is capable of producing outputs that show transition between states in response to input (Supplementary Fig. 7A). Specifically, through the influence of random pulses from the inputs, the network learns to maintain each state and transition to the next based on the incoming input.

To observe how the trained network switches between memory states, we focus on a specific single transition (i.e., from a memory state $(-1, -1, -1)$ to another one $(1, -1, -1)$). We perturbed the state of the network with input pulses to observe transitions between two fixed points over six trials. In these trials, the strength of the input was gradually increased. We visualized the network states in a state space spanned by the first principal component vector and the input vector in order to observe both the effects of input perturbations and memory transitions. When the input was weak, we found that the network activity briefly deviated from its fixed point and, when the input was removed, returned to the original fixed point. In striking contrast, when the input was strong enough, the network switched to a different fixed point, suggesting the existence of saddle nodes between the two stable fixed points (Supplementary Fig. 7B).

In summary, these results suggest that the predictive alignment can learn multiple stable fixed point attractors. Importantly, the learned stable fixed-point attractors were separated by saddle nodes that provide transitions between states, and such saddle nodes were not explicitly learned.

## Learning low-dimensional chaotic attractor: the Lorenz attractor

The tasks we have shown so far are relatively simple, with periodic targets only. This raises the question of whether our models can learn more complex dynamics such as the three-dimensional Lorenz attractor (shown in Fig. 4G) as a target trajectory. Unlike the simple periodic signals studied in the previous scenarios, the Lorenz attractor exhibits non-periodic and complex behavior.

We considered a network with three readout units to learn the three-dimensional Lorenz attractor. Even though the desired targets are not periodic, the late phase of learning showed readout dynamics that closely matched the desired output (Fig. 4A, 0–3 s). We found that, when plasticity was off, the output predicted the subsequent dynamics well, but then showed divergence from the targets because the model itself is a chaotic system. Interestingly, despite this deviation from the desired dynamics, the readout activity still showed complex oscillations similar to those of the target.

We compared the trajectories of the dynamics of the trained network when plasticity was off and the target signals. Both the projected trajectories on the two-dimensional planes and the full trajectories show striking similarities between the two, suggesting that the

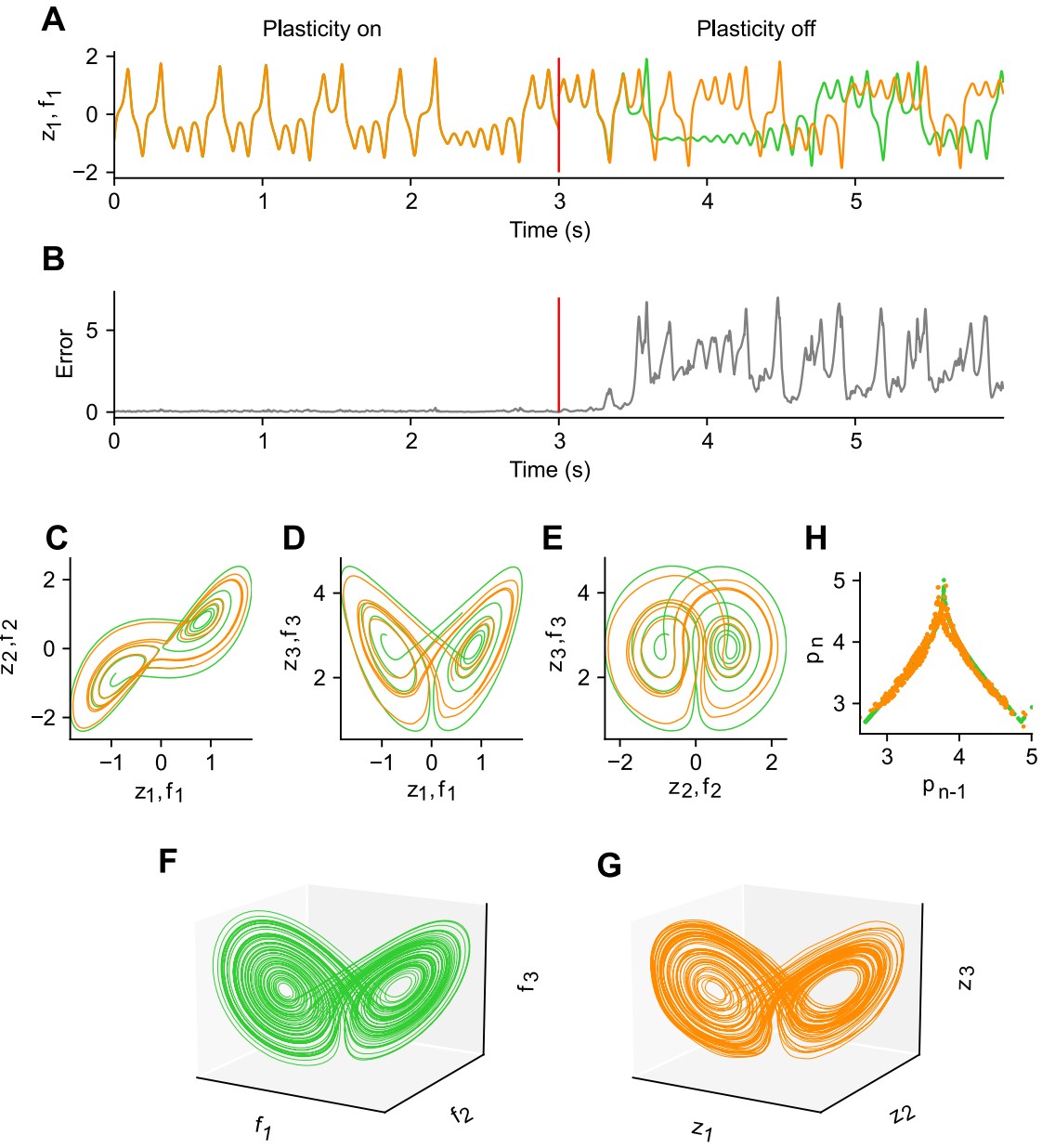

**Fig. 4 | Learning chaotic dynamics: the Lorenz attractor. A** First component of target Lorenz attractor signal (green) and the output activity of the network (orange). The vertical red line represents the time when plasticity was turned off from on after learning. The output of the network after training showed almost the same dynamics as the target signal when the plasticity was on. Even after the plasticity was turned off, the output initially showed the same dynamics as the target, but after a while it showed a pattern different from the target but similar to Lorenz dynamics. **B** Error between the two signals shown in (**A**) is shown. **C**–**E** Trajectories projected on two-dimensional state spaces are shown. **F** Target Lorenz attractor in three-dimensional state space is shown. **G** Same as in (**F**) but for the outputs. **H** Tent map representation plotting the current local maximum of the third component against the previous local maximum for the target signal (green) and network output (orange) is shown.

model learns the manifold representative of the true Lorenz attractor (Fig. 4C–G). Quantitative analysis of the tent map[29] of successive maxima relative to the previous maxima further indicates that the model has adequately learned the Lorenz attractor (Fig. 4H).

In summary, we have shown that predictive alignment can learn and generate low-dimensional complex autonomous dynamics through recurrent plasticity.

### Learning generalized representations

The above results have demonstrated that the predictive alignment can learn and generate complex target signals by adapting both readout and recurrent connections through plasticity. This was achieved by precise adjustment of both recurrent and output weights. In a machine learning framework called Reservoir Computing (RC), it has been shown that recurrent networks can learn complex tasks without relying on the plasticity of recurrent connections[13–20]. This is achieved by generating rich basis function in the network dynamics such that the readout unit can decode the arbitrary output signals. Inspired such a rich generalization ability in the recurrent networks, we then probed the model's potential to generalize pretrained simpler output patterns to more complex targets without recurrent plasticity.

We considered a structured task with a network consisting of two different groups of readout units to answer our above question. The weights projecting to these two groups of readout units and the

recurrent weights were learned differently through two distinct learning phases. The first group underwent the predictive alignment learning to generate a set of simple multi-frequency sinusoidal signals (Supplementary Fig. 8A; left). During this first learning phase, both recurrent and readout weights were trained on the first group of readout units, while the weights on the second group of readouts remained static. Once the first group was successfully trained to generate sinusoidal targets, we introduced a second phase into the learning process, in which recurrent plasticity was terminated while the readout weights of the second group were started to train to generate novel target signals (Supplementary Fig. 8A; right). We basically used our pretrained network with the predictive alignment as a reservoir computing network, in terms of learning only readout weights with fixed recurrent weights. Note that, in principle, any periodic function can be approximated by a summation of a sufficient number of sinusoidal components. The second group successfully learned and generated novel signals, despite the fact that the recurrent connections were not trained to optimize for such targets (Supplementary Fig. 8B). In contrast, the network without training in the first phase failed to learn the target signals in the second phase, as the network did not learn the sinusoids (Supplementary Fig. 8C).

In summary, we have shown that the network can generalize from learning simple multi-frequency oscillations to learn and generate novel signals without relying on the plasticity of recurrent connections.

## Learning Ready-Set-Go task

Up to now, we have explored whether the network can learn autonomous dynamics or input-output transformations. We then wondered whether the proposed mechanism could learn more complex task, Ready-Set-Go (RSG), in which the desired output was indicated by the time interval between two identical pulse inputs. In the RSG task, the network was required to measure that interval, keep it in memory during a delay period of random duration, and then reproduce it after the set signal[30]. This task is more difficult than the production of a periodic output due to the requirement for the RNN to learn to store the information about the interpulse delay, and then produce responses at different times depending on the value of delay.

To mimic such an RSG in our simulation, we considered two input units sending pulses to the network. In each trial, one of four delay

values $T_{delay}$ was sampled and the two units generated pulses with an interpulse delay of $T_{delay}$ (Fig. 5A). Note that our delay values were more than ten times larger than the time scale of each neuron (i.e., 10 ms). The network projected to a single readout unit and the weights were modified such that the output should be a pulse delayed by $T_{delay}$ relative to the second input pulse. After learning, network generated the output with the desired time delay over all four samples (Fig. 5B; colored squares). This generation of the desired timing results from the phasic responses of the network activity, which are controlled by the time delay between input pulses (Supplementary Fig. 9). More interestingly, the network interpolated well to input intervals in between those used for training, while failed at extrapolation (Fig. 5B; gray curve, Fig. 5C). These results suggest that the network did not simply learn individual output mappings, but instead learned an underlying manifold structure encoding the range of possible delays.

We then wondered whether the low dimensional network dynamics already shows the structured representations for the temporal delay information in the task. To test this prediction, we performed the principal component analysis (PCA) on the trained network and visualized the low dimensional representation in the network. Projected dynamics on the two-dimensional PC axis showed linear shifts of trajectories along the linear manifold with increasing delay, revealing that the temporal structure of the task is embedded in the low-dimensional recurrent dynamics (Fig. 5D). Taken together, these results demonstrate the network's ability to generalize by acquiring the underlying manifold structure inherent to the task.

## Learning with spiking recurrent network

To test the applicability of predictive alignment to spiking recurrent networks (SRN), we implemented a recurrent network consisting of 1,000 leaky integrate-and-fire (LIF) neurons coupled to a single linear readout unit as the output (see Methods). We found that, as in the rate-based network model, the readout unit of the spiking model learned to generate a smooth sinusoidal waveform that closely matched the target signal (Supplementary Fig. 10A), the performance of which was compatible with FORCE and e-prop (Supplementary Fig. 10C). Furthermore, we showed that the proposed learning rule enables learning even in a recurrent spiking neural network composed of two populations (i.e., excitatory and inhibitory) (Supplementary Fig. 11).

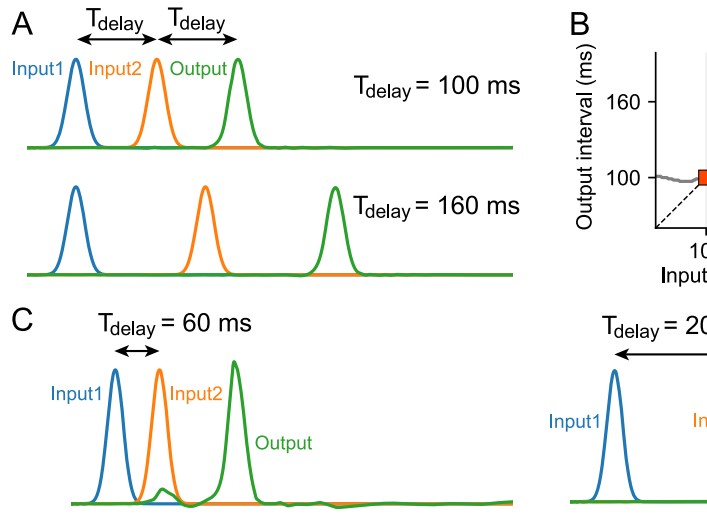

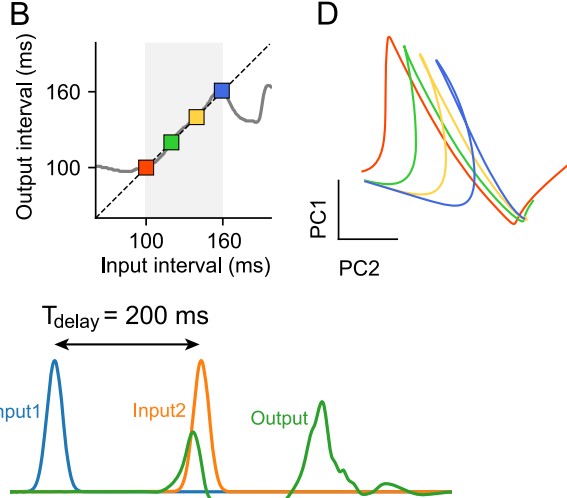

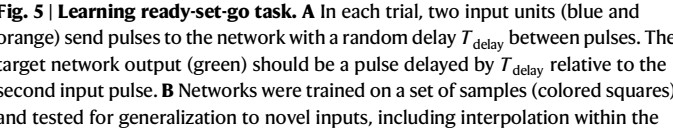

**Fig. 5 | Learning ready-set-go task. A** In each trial, two input units (blue and orange) send pulses to the network with a random delay $T_{delay}$ between pulses. The target network output (green) should be a pulse delayed by $T_{delay}$ relative to the second input pulse. **B** Networks were trained on a set of samples (colored squares) and tested for generalization to novel inputs, including interpolation within the training range (shaded region) as well as extrapolation beyond the training range. **C** The network after training failed to extrapolate beyond the training range. **D** Principal component analysis (PCA) of the trained network revealed that as the delay $T_{delay}$ increased, the network states corresponding to the output peak shifted linearly along a manifold in state space.

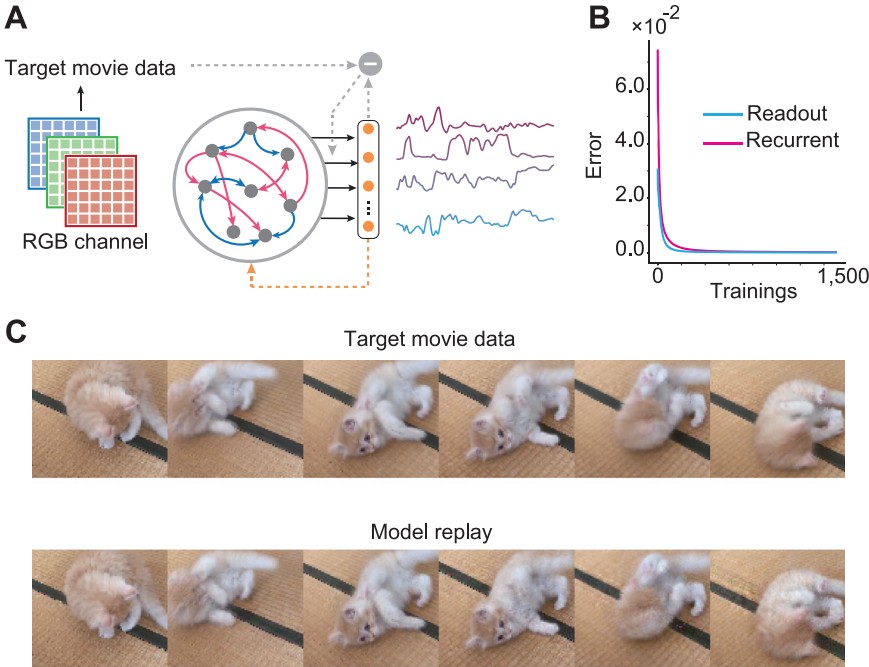

**Fig. 6 | Predictive alignment for movie data storage and replay. A** Schematic of the network architecture for learning video sequences, with 22,080 pixel target signals and 800 units in the recurrent network. **B** Learning curves showing the monotonic decrease in readout weight error (cyan) and recurrent weight change (magenta) over 1500 epochs of training on the upsampled 1000-frame video sequence. **C** Samples of the original video frames (top) compared to the output frames generated by the network (bottom) after training.

## Learning and replay of high-dimensional movie data

Finally, we demonstrate that the model can learn natural high-dimensional signals. Here, we train recurrent neural network to learn and replay high-dimensional video patterns in pixel space. The original video sequence consisted of 100 frames. However, to enable learning of finer temporal dynamics, this sequence was temporally upsampled by interpolation to a higher resolution of 1,000 frames for use during training. Each individual video frame had a spatial dimension of 80×92 pixels, with three color channels (RGB) per pixel, resulting in a total of 22,080 units representing the pixel values (Fig. 6A). In contrast to the high-dimensional target signals, the recurrent neural network model itself consisted of only 800 units. During training, weights were modified to minimize the error between 22,080 readout activities and the movie data. At time zero in each trial, we set the network activity to the predefined initial states, generated by Gaussian distributions with zero-mean and the standard deviation of one. The same initial state was used in both the learning and testing phases.

Despite the large dimensionality mismatch between the target signals and the recurrent layer, the readout and recurrent weight training errors decreased monotonically and converged to near zero values after sufficient training iterations, suggesting the network dynamics were able to be trained successfully (Fig. 6B). Indeed, readout activities driven by the autonomous network dynamics after learning showed that the network dynamics accurately encoded and replayed the full training video patterns (Fig. 6C; Supplementary Movie 1). These results indicate that the proposed predictive alignment is not restricted to few target signals but can be applicable to high dimensional signals.

## Discussion

Many cortical and subcortical circuits exhibit complex, dynamic patterns of activity that underlie crucial cognitive functions such as working memory[31], decision-making[32,33], motor control[34], and sensory processing[35]. In this paper, we have presented a novel learning framework which we call "predictive alignment" that enables recurrent neural networks to learn and generate a variety of patterned activities, even when the network initially exhibits chaotic spontaneous dynamics. The key insight of our approach is that instead of directly minimizing the output error to train recurrent connections, as in most existing recurrent learning rules[2,28,36–39], we instead focus on predicting the recurrent activity itself and aligning this prediction with the chaotic dynamics of the network. This allows us to efficiently suppress the chaotic spontaneous activity and shape the network dynamics to produce the desired patterns.

The predictive alignment learning rule we proposed has several advantages in terms of biological plausibility. First, the synaptic plasticity rules are local, depending only on the activity of the pre- and post-synaptic units and their predicted activities, without requiring non-local information such as the inverse of the correlation matrix of the network activities[21] or unfolding the dynamics through time[36,37]. Second, our framework does not rely on clamping the network's outputs to target signals. FORCE learning and its variants[10,21,40–43] tamed the chaos by clamping the output to be close to the target during learning. They use weight changes which are faster than the time scale of the dynamics, which seems to lack biological plausibility. Instead, our predictive alignment learns to generate the desired patterns by predicting the feedback signal with aligning the recurrent and the chaotic dynamics, which does not require such assumptions. Since our rule uses multiple localized activities within each unit (e.g., $Mr$ and $Gr$), we expect that multicompartmental recordings from individual neurons during a learning task would enable to test whether biological neural circuits use similar local predictive learning rules. The proposed rule predicts that the correlation between multiple dendritic synaptic currents and/or somatic membrane potentials should increase as learning progresses. Further experimental studies are needed to better understand how such rules might give rise to structured neural dynamics.

Recently, several local learning rules for recurrent neural networks have been proposed, such as FOLLOW[22] and RFLO[23]. While the FOLLOW rule relies on clamping outputs to target values similar to FORCE, it

achieves this clamping via negative error feedback from an auto-encoder, allowing slower weight changes that make the error signal locally available at post-synaptic units. In contrast, our model does not require clamping at all because it can learn to generate desired output patterns even when the initial recurrent dynamics differ from the target, as described above. Furthermore, because FOLLOW drives outputs based on input dynamics, it cannot perform the delayed adaptation tasks considered here[23]. Another framework, RFLO, approximates backpropagating output errors through random feedback connections to train the recurrent connections[23]. While both our predictive alignment and the RFLO rely on random feedback projection from the output, the two rules have significant difference: the RFLO feedback the output error signal, whereas our rule adapts recurrent predictions to the output signals directly without relying on output errors.

Another powerful learning framework that approximates Back-propagation Through Time (BPTT)[36,37], which is the most celebrated training method for recurrent neural networks in machine learning, is the e-prop[24]. In the e-prop, the gradual changes in the hidden variables of units generate eligibility traces that extend over long periods of time which in turn are combined with subsequent instantaneous error signals. While our model also combines the instantaneous error signal and the presynaptic activity, there are some differences between the two. First, although the e-prop assumes low-passed presynaptic activity as an eligibility trace, our rule considers instantaneous presynaptic activity. Second, our rule requires the instantaneous error between the output activity via the feedback signal $\mathbf{Qz}$ and the regularized recurrent prediction $\hat{\mathbf{j}}\mathbf{r}$, whereas the e-prop utilizes the error between the desired target signal and the output activity. Future work should clarify whether the proposed rule can approximate BPTT in some way, as e-prop does.

How can a predictive alignment framework be realistically implemented in the brain? The motor cortex provides a prime example of a recurrent circuit capable of learning complex temporal patterns. Neurons in the motor cortex receive input about the planned action from the premotor cortex and information about the current state of the body from sensory input. They are thought to use this combined signal to compute the error necessary for learning and adjusting their neural activity patterns to generate the desired output trajectory[44]. In the predictive alignment framework, instead of transmitting an error signal derived from a target output, the feedback from the network's own output could be transmitted to each unit as a "teacher" signal. Previous experimental studies suggest that the apical dendritic compartment receives input from higher cortical layers, which in turn is attenuated before reaching the somatic compartment[45]. Our framework predicts that this attenuated apical input does not directly influence somatic firing rates, but instead acts as a teaching signal that incorporates predictive feedback, allowing individual neurons to modify their recurrent connections to better anticipate and generate the required motor outputs. Further experimental studies would be needed to understand the specific biological mechanisms that could organize such a predictive alignment process within recurrent cortical circuits.

Our plasticity rule assumes two types of recurrent connections (i.e., $\mathbf{M}$ and $\mathbf{G}$), one is plastic and the other is static. While the biological relevance of this is still an open question, it has been reported experimentally that the degree of synaptic plasticity varies across different compartments of dendrites[46]. Based on this experimental evidence, we can speculate that the plastic connections could be synapses that project onto proximal basal dendrites and the static connections could be projecting onto distal basal dendrites.

We considered the predictive alignment in a recurrent network with balanced excitatory and inhibitory synaptic currents. The implementation we used in this paper does not satisfy Dale's law[47]. A more biologically plausible model would require separate excitatory and inhibitory neuron populations[48]. It would be an interesting question how maintaining an appropriate excitation/inhibition

balance via inhibitory plasticity[49–52] is critical for stable learning in such two-population network with chaotic spontaneous activity. Future extensions of our model should explore more biologically plausible architectures consists of spiking neurons as well[41,53–58]. Another interesting direction would be to explore how our predictive plasticity interacts with reward signals. Reward signals could potentially guide the shaping of top-down feedback signals[59], with predictive alignment then fine-tuning the dynamics. Alternatively, predictive and reward-based plasticity could operate in parallel on distinct synaptic subpopulations. Combining these plasticity mechanisms could provide new insights how different forms of learning interact to give rise to robust and flexible neural computation in reinforcement learning.

In conclusion, predictive alignment presents a novel and biologically plausible learning approach for training chaotic recurrent neural networks. This framework suggests that prediction within the local circuit can guide powerful, robust and generalizable learning.

## Methods

### Predictive alignment learning rule

Our network consists of $N$ rate-based units, mutually connected with the recurrent connections. Here, we considered two types of recurrent connections: strong and fixed connectivity $\mathbf{G}$ and initially weak yet plastic connectivity $\mathbf{M}$. The strength of each connection was generated by a Gaussian distribution, and unless other specified, with zero-mean and the standard deviation of $g/\sqrt{pN}$, with $(p,g)=(0.1,1.2)$ for $\mathbf{G}$ and $(p,g)=(1.0,0.5)$ for $\mathbf{M}$. Here, $g$ is a gain of connection, leading chaotic spontaneous activity with $g > 1$, thus the fixed recurrent connectivity $\mathbf{G}$ generates initial chaotic spontaneous activity. The dynamics of membrane potential with the external input $\mathbf{I}$ was governed by the following equation:

$$\tau\dot{\mathbf{x}} = -\mathbf{x} + (\mathbf{G}+\mathbf{M})\mathbf{r} + \mathbf{W}^{\text{in}}\mathbf{I} + \sigma\boldsymbol{\xi} \qquad (8)$$

where $\boldsymbol{\xi}$ is the time-varying Gaussian noise with zero mean and unit variance, and the strength of noise was controlled by the scalar value of $\sigma$. We have controlled the noise level in Supplementary Fig. 3, while setting it to zero for the rest of our simulations. The variable $\mathbf{x}$ is a membrane potential and $\mathbf{r}$ is a firing rate of unit, defined as:

$$\mathbf{r} = \phi(\mathbf{x}) \qquad (9)$$

$$\phi(\mathbf{x}) = \tanh(\mathbf{x}) \qquad (10)$$

The matrix $\mathbf{W}^{\text{in}}$ is a feedforward connectivity projecting from input layer, which is assumed to be fixed during the whole simulation. The parameter $\tau$ is a membrane time constant, which we set as $\tau = 10$ (ms). The recurrent units project to readout units, of which the output value was defined as a linear summation of firing rates:

$$\mathbf{z} = \mathbf{W}\mathbf{r} \qquad (11)$$

where $\mathbf{W}$ is a readout weight matrix.

The cost function for the plastic recurrent connectivity is defined over a period of learning phase $T$ as follows:

$$\mathcal{L}_{\text{rec}} = \frac{1}{2T}\int dt ||\mathbf{Qz} - \mathbf{Mr}||^2 - \frac{\alpha}{T}\langle\mathbf{Gr}, \mathbf{Mr}\rangle \qquad (12)$$

where $\mathbf{Q} \in \mathbb{R}^{N \times K}$ is a random feedback matrix projecting from the readouts and $\langle\mathbf{Gr}, \mathbf{Mr}\rangle$ is a correlation defined as:

$$\langle\mathbf{Gr}, \mathbf{Mr}\rangle = \int dt(\mathbf{Gr})^{\mathsf{T}}(\mathbf{Mr}) \qquad (13)$$

All elements of $\mathbf{Q}$ were chosen randomly and uniformly over the range $-3/\sqrt{K}$ to $3/\sqrt{K}$, where $K$ is a number of readouts. Taking the gradient of this cost function, we obtain an online learning rule as:

$$\Delta\mathbf{M} = -\eta_M \frac{\partial\mathcal{L}_{rec}}{\partial\mathbf{M}} = \eta_M\{[\mathbf{Qz} - \mathbf{Mr}]\mathbf{r}^T + \alpha\mathbf{Grr}^T\} = \eta_M[\mathbf{Qz} - (\mathbf{M} - \alpha\mathbf{G})\mathbf{r}]\mathbf{r}^T$$

(14)

where T is a transpose. Readout weights were trained by simple least-mean-square (LMS):

$$\Delta\mathbf{W} = \eta_W(\mathbf{f} - \mathbf{z})\mathbf{r}^T$$

(15)

where $\mathbf{f}$ is a vector of teaching signal for the model's outputs.

## Details for learning Lorenz attractor

In Fig. 4, we trained a network with three readout units with Lorenz attractor as the target signal following the dynamics below:

$$\dot{f}'_1 = s(x_2 - x_1)$$
$$\dot{f}'_2 = rx_1 - x_2 - x_1x_3$$
$$\dot{f}'_3 = x_1x_2 - bx_3,$$
$$f_k = f'_k/10 (k = 1, 2, 3)$$

(16)

where $(s, r, b) = (10, 28, 8/3)$ in our simulation. Learning was performed using a 15,000 s trajectory generated according to these dynamics as a target signal.

## Details for ready-set-go task

In Fig. 5, we trained a network consists of $N = 1,200$ neurons to learn the Ready-Set-Go task. The two input pulses ($s_1$ and $s_2$) and a target pulse ($o$) were scaled Gaussians with a standard deviation of $\Delta = 15$ ms. The two inputs had a delay $T_{delay}$ between them. All pulses had a uniform shift of $T_0 = 60$ ms to ensure that the first pulse was presented after the start of each trial. The network was trained for 200,000 trials, and in each trial, the delay value $T_{delay}$ was randomly selected from one of the four values (100 ms, 120 ms, 140 ms, and 160 ms).

$$s_1(t) = 2\exp\left[-\frac{(t - T_0)^2}{\Delta^2}\right] - 1$$

(17)

$$s_2(t) = 2\exp\left[-\frac{\left(t - T_0 - T_{delay}\right)^2}{\Delta^2}\right] - 1$$

(18)

$$o(t) = 2\exp\left[-\frac{\left(t - T_0 - 2T_{delay}\right)^2}{\Delta^2}\right] - 1$$

(19)

## Details for learning with noise

In Supplementary Fig. 2, we trained a network with various levels of noise by changing the scalar value of $\sigma$ (see Eq. 7). In all conditions, we trained the network with the predictive alignment and the FORCE for 100 s. To train with FORCE, we chose the regularization parameter of unity, which is the standard setting in FORCE learning[21].

## Details for learning fixed-point attractors

In Supplementary Fig. 7, the neural network was trained using randomly presented pulses as inputs. The strength of each pulse during training was either +1 or −1, with a duration of 100 ms. The intervals

between consecutive pulses varied randomly within a range of 500−700 ms. The network was trained for 300 s. After training, the network was stimulated with similar pulses while varying the intensity between 0.27 and 0.33.

## Quantification of representational diversity and information balance

In Supplementary Fig. 6, to assess the network's ability to generate diverse and structured representations, we analyzed the correlation matrix $C$ of the network activity. The entropy of the eigenvalues $H_\lambda$ was computed as

$$H_\lambda = -\sum_i p_i \log p_i, p_i = \frac{\lambda_i}{\sum_j \lambda_j}$$

(20)

where $\lambda_i$ are the eigenvalues of $C$, and $p_i$ represents their normalized contributions. This measure quantifies how evenly the network's activity is distributed across different representational modes. Note that we analyzed the network dynamics before training.

To evaluate the effective dimensionality of the network's activity, we computed the participation ratio:

$$PR = \frac{\left(\sum_i \lambda_i\right)^2}{\sum_i \lambda_i^2}$$

(21)

which reflects the number of significant dimensions contributing to the network's representations. To prevent excessive weighting of high-dimensional representations and to maintain a more proportional scaling with the network's effective dimensionality, we applied a square root transformation to the participation ratio as a measure of representational balance. Finally, we defined a efficiency index as:

$$\text{Efficiency} = \frac{H_\lambda}{\sqrt{PR}}$$

(22)

and examined how this metric varied across different network regimes. We found that this index was maximized at the edge of chaos, indicating that learning performance is optimal when representational diversity and effective dimensionality are properly balanced.

## Details for estimating Lyapunov exponent

In Supplementary Fig. 4, we quantified the degree of chaos using the Lyapunov exponent $\lambda^*$, defined as follows.

$$\lambda^* = \lim_{k\to\infty}\frac{1}{k}\log\left(\frac{\gamma_k}{\gamma_0}\right)$$

(23)

with $\gamma_0$ being the initial distance between the perturbed and the unperturbed trajectory, and $\gamma_k$ being the distance at time $k$. For subcritical systems, $\lambda^* < 0$ and for chaotic systems $\lambda^* > 0$.

At each training step, we generated two copies of the network with states denoted as $x^{(1)}$ and $x^{(2)}$. We introduce a small perturbation to initially separate the states by $\gamma_0 = 10^{-6}$. Next, we simulated the network for 500 steps without inputs to eliminate transient effects from the initial states. After this initialization, we ran the network for 1,000 additional steps—also without inputs—to estimate the Lyapunov exponent. During this process, we recorded the state difference at each step, defined as $\gamma_k = ||x^{(1)}(k) - x^{(2)}(k)||$. To prevent the difference between $x^{(1)}$ and $x^{(2)}$ from saturating or diverging, we reset the distance to $\gamma_0$ at every step while preserving the direction of the vector $x^{(2)}(k) - x^{(1)}(k)$, as follows[60]:

$$x^{(2)}(k) \leftarrow x^{(1)}(k) + \frac{\gamma_0}{\gamma_k}\left(x^{(2)}(k) - x^{(1)}(k)\right)$$

(24)

Then, the average of the logarithm of the distances was calculated as follows:

$$\lambda = \left\langle \log\left(\frac{\gamma_k}{\gamma_0}\right) \right\rangle_k \qquad (25)$$

### Details for learning spiking recurrent network

In Supplementary Fig. 10, we trained a network consisted of 1,000 leaky integrate-and-fire (LIF) neurons, coupled with a single linear readout unit as the output. The network obeyed the following dynamics:

$$\tau_m \dot{\mathbf{v}} = -\mathbf{v} + (\mathbf{G} + \mathbf{M})\mathbf{r} + \mathbf{I}_{bias} \qquad (26)$$

where $\tau_m$ is a membrane time constant of 10 ms and $\mathbf{v}$ is a membrane potential over a whole network. Once the membrane potential exceeds a threshold of $v_{th} = 1$, it was reset to a resting potential of $v_{reset} = -1$. The LIF model has a refractory period of $\tau_{ref} = 2$ ms, where the neuron cannot generate spike. The vector $\mathbf{I}_{bias}$ is a constant background current set at the threshold value. Each element of the synaptic current vector $\mathbf{r}$ is a filtered spike train as:

$$\dot{r}_i = -\frac{I}{\tau_s} + \frac{1}{\tau_s}\sum_{t_{ij}} \delta\left(t - t_{ij}\right) \qquad (27)$$

where $\tau_s = 20$ ms is the time constant for the filter and $t_{ij}$ is the $j$-th spike fired by the $i$-th neuron. The function $\delta(\cdot)$ is the Dirac delta function. The output of the network was calculated as a linear summation of synaptic currents with the readout weights, just as in the rate-based network (i.e., Eq. 10).

Similar to our rate-based model, we considered two types of recurrent connections: strong and fixed connectivity $\mathbf{G}$ and initially weak yet plastic connectivity $\mathbf{M}$. The strength of each connection was generated by a Gaussian distribution, with zero-mean and the standard deviation of $g/\sqrt{pN}$, with $(p, g) = (0.1, 0.3)$ for $\mathbf{G}$ and $(p, g) = (1.0, 0.05)$ for $\mathbf{M}$. Further, the mean for the realization for these weight matrices was explicitly set to 0 to counterbalance the resulting firing rate heterogeneity[42]. All weights were trained with the same plasticity rules in the rate-based network (i.e., Eqs. 13 and 14).

In Supplementary Fig. 10C, we compared learning performance across different spiking models: predictive alignment, FORCE[42], and e-prop[24]. All models were composed of Leaky Integrate-and-Fire neurons. For all models, we used a network of 500 neurons and trained on a 5 Hz sine wave for 1000 epochs. In e-prop, we used 100 input neurons, generating frozen Poisson spiking patterns with the instantaneous firing rate of 50 Hz. Learning accuracy was evaluated using the root mean square error (RMSE) at the final epoch. The results are presented as a bar graph, with error bars indicating the standard deviation.

In the two-population scenario with the distinct excitatory and inhibitory populations,

$$\tau_m \dot{\mathbf{v}}_E = -\mathbf{v}_E + (\mathbf{G}_{EE} + \mathbf{M}_{EE})\mathbf{r}_E - (\mathbf{G}_{EI} + \mathbf{M}_{EI})\mathbf{r}_I + \mathbf{I}_{bias}^E \qquad (28)$$

$$\tau_m \dot{\mathbf{v}}_I = -\mathbf{v}_I + (\mathbf{G}_{IE} + \mathbf{M}_{IE})\mathbf{r}_E - (\mathbf{G}_{II} + \mathbf{M}_{II})\mathbf{r}_I + \mathbf{I}_{bias}^I \qquad (29)$$

with the fixed connections $\mathbf{G}_{XY}$ and the plastic connections $\mathbf{M}_{XY}$, where X and Y indicate either excitatory or inhibitory. All connections were first generated using a Gaussian distribution, as in the single population case, and then any negative values were flipped to positive. Both populations were assumed to have connections to the readout unit.

### Simulation details

All simulations were performed in customized Python3 code written by TA with numpy 1.17.3 and scipy 0.18. Differential equations were numerically integrated using a Euler method with integration time steps of 0.5 ms in Supplementary Fig. 8 and 1 ms in the rest of the simulations. All source code will be available after publication on GitHub and will be distributed to the reviewers.

### Reporting summary

Further information on research design is available in the Nature Portfolio Reporting Summary linked to this article.

## Data availability

All numerical datasets necessary to replicate the results shown in this article can easily be generated by numerical simulations with the software code provided below. No datasets were generated during this study.

## Code availability

All codes were written in Python3 with numpy 1.23.5 and scipy 1.10.1. Example program codes used for the present numerical simulations and data analysis will available at: https://github.com/TAsabuki/PredictiveAlignment[61].

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

## Acknowledgements

The authors express their sincere thanks to Guillaume Bellec for his valuable comments on our manuscript. This work was supported by BBSRC BB/N013956/1, BB/N019008/1, Wellcome Trust 200790/Z/16/Z, Simons Foundation 564408 and EPSRC EP/R035806/1.

## Author contributions

T.A. and C.C. conceived the study and wrote the paper. T.A. performed the simulations and data analyses.

## Competing interests

The authors declare no competing interests.
