## [Transparent Peer Review file · Nature Communications]

Taming the chaos gently: a Predictive Alignment learning rule in recurrent neural networks

Corresponding Author: Dr Toshitake Asabuki

A version of this paper was originally rejected for publication by Nature Communications, however that decision was reconsidered after appeal by the authors.

Version 0:

Reviewer comments:

Reviewer #1

(Remarks to the Author)

In this manuscript, the authors present a learning method for firing rate-based recurrent neural networks (RNNs), referred to as predictive alignment, and demonstrate that the method can learn and generate a variety of patterned activities, ranging from periodic signals to chaotic Lorenz attractors and high-dimensional movie data.

Strengths

As neural circuits exhibit complex chaotic spontaneous activity, understanding how such chaotic activity can be efficiently reorganized to give rise to coherent activity patterns that generate behaviour is indeed an important question. The study attempts to address this question, particularly from the angle that prediction, to some extent, can guide learning.

Weaknesses

This study, however, has some serious weaknesses, indicating that the results reported in this manuscript do not necessarily represent a significant advance in the field. More on these in a point-by-point fashion below:

(1) From a machine learning point of view, the tasks that can be performed by training RNNs using the predictive alignment method can all be implemented by the FORCE method and its variants, often with better performance; some of these papers are cited in the manuscript. From a biological perspective, the methods developed in this study lack the biological plausibility of training RNNs with spiking neurons. Regarding this latter point, the e-prop method (Bellec et al., Nature Communications, 2020), which is also local and leverages error signals, offers more biological plausibility. As the field stands now, a powerful new method that can effectively train recurrent spiking neural networks would represent significant progress.

(2) It is unclear why the learning method works. In other words, what changes occur during the learning process that enable the RNNs to perform the intended computations? Are these changes related to the reorganization of the underlying chaotic attractor into a rich combination of fixed points or continuous attractors of the corresponding dynamical systems? I would like to see a deeper treatment of this question to open the "black box" of this process. The learning performance is best at the "edge of chaos," which is a very interesting observation, but why do such critical dynamics facilitate learning?

(3) What is the biological relevance of assuming the recurrent connectivity is a summation of M and G ?

(4) What are the effects of certain hyperparameters, such as learning rates, on learning performance? Without thorough exploration of this aspect, it is difficult to assess whether the learning method is robust and effective.

(Remarks on code availability)

Reviewer #2

(Remarks to the Author)

Here the authors propose a novel learning rule to “tame” high-dimensional chaotic patterns in recurrent neural networks. The learning rule, “predictive alignment”, adjusts recurrent weights in order to best predict the feedback in an initially chaotic RNN network. After training, the chaotic activity resembles an RNN in which the network is suppressed by an external or feedback signal except that the suppression is driven by the recurrent network (although by an independent recurrent weight matrix M). The authors show that RNNs trained with predictive alignment can learn a number of easy and difficult time-varying tasks. An important contribution of this paper is that the learning rule is indeed much more biologically plausible in terms of the learning rule itself. However, there are other aspects of the implementation that are not normally present in RNN models and might be considered implausible by some (e.g., the “silent feedback” and the two weight matrices). But these are not necessarily implausible, and may be considered to be predictions. Overall the presentation of the novel and potentially biologically plausible model is a significant contribution, however, the methods are very brief and the task implementations are not well described, making it somewhat difficult to evaluate the work in detail.

My primary concern relates to the apparent absence of noise in the RNN. As far as I could tell by looking at the Methods and sample code (perhaps I’m wrong), in contrast to most RNN models, there was no noise term during training or testing. When proposing a biologically plausible learning rule that relies on feedback from a chaotic RNN the absence of noise is potentially a serious issue, because training may not work at all when noise is present as the feedback signal is a moving target. It is thus necessary to include noise and parametrically vary noise to understand the influence of noise in predictive alignment.

The use of the Measure-Wait-Go task is interesting, and an important demonstration of the power of the learning rule. However, the time scale used was less than an order of magnitude longer than that of τ . Does the approach work for more biologically relevant time scales such as those used in the Jazayeri and Shadlen paper: hundreds of milliseconds? Also, this task is normally referred to as the ready-set-go task (Jazayeri and Shadlen, 2010). It would also be helpful to show a sample of the RNN activity during the task rather than just the PC’s, because linear PCs can emerge from RNNs without much linear activity.

Replay task. It was not clear what time = 0 was for the movie generation? Does the produced movie run in a limit cycle?

Pg. 2. To support the statement “Cortical circuits often exhibit chaotic spontaneous activity” a number of papers are cited, but they all seem to be models, not experimental papers (and I don’t think Rajan et al, 2016 even address chaos). RNNs are not cortical circuits so these references do not seem appropriate. London, ..., Latham, 2010 provides some evidence for chaos in cortical circuits.

The presentation and discussion omitted many relevant studies that laid the foundation for the current work.

Jaeger and Haas 2004, essentially published what later came to known as FORCE in their “online” learning rule.

The paper has commonalities with Laje and Buonomano, which also took the approach of taming chaos by changing the recurrent weights and using high dimensional structure of a local stable trajectories as a reservoir. That work was not cited in the text. And Buonomano and Merzenich 1995, was the first description of what later came to be called a reservoir network.

Supplemental Figure 4. A schematic of the network would be helpful.

Pg. 14 “while failed AT extrapolation”

Pg. 15 “Here, we train recurrent neural” ... typo.

It is best to refer to the units and weights as such, rather than neurons and synapses.

(Remarks on code availability)

Version 1:

Reviewer comments:

Reviewer #1

(Remarks to the Author)

I appreciate the authors' detailed responses and the corresponding revisions. However, my concerns regarding the training of RNNs with spiking neurons and the changes that occur during the learning process to enable these networks to perform the intended computations remain unclear. Specifically:

(1) For the proposed learning rule to be effectively evaluated in a spiking neural network, it is essential to compare its performance with existing methods for training SNNs. Additionally, applying the proposed method to balanced spiking neural networks with both excitatory and inhibitory neurons would be more appropriate for two reasons: (i) it aligns better with biological plausibility, and (ii) balanced networks are known to exhibit chaotic dynamics, which are more consistent with the problem addressed in this study.

(2) While the revision presents some results on learned fixed points, it remains unclear how and why the underlying chaotic dynamics are reorganized during the training process.

(3) The added explanation for why the model performs best at the edge of chaos is largely qualitative and descriptive (see Line 310). In machine learning, it is well established that artificial neural networks achieve optimal training performance near the edge of chaos [Schoenholz, et al., Deep information propagation, ICLR 2017], supported by quantitative analysis. Similarly, I believe a quantitative investigation is necessary to strengthen this argument.

(Remarks on code availability)

Reviewer #2

(Remarks to the Author)

The authors have adequately addressed my concerns, and the inclusion of the spiking novel further improves the paper?

"matrixes" should be "matrices"

(Remarks on code availability)

Version 2:

Reviewer comments:

Reviewer #1

(Remarks to the Author)

In the revised manuscript, the authors have included comparisons between the proposed learning rule and two other supervised learning methods, and have added some quantitative explanations concerning why learning performance is maximised at the edge of chaos. My main concerns have now been addressed in the revisions.

(Remarks on code availability)

Response to the reviewers

Toshitake Asabuki and Claudia Clopath

We thank the reviewers for their thorough and critical evaluation of our work. We read the reviewers' comments carefully and we **think we have addressed reviewers' concerns fully**. Below, we explain how we addressed each of their concerns point-by-point. In particular, our additional results show that we can implement our rule in a spiking network and that the rule is robust to noise, two majors concerns of the reviewers. Thank you very much in advance for your time.

Reviewer #1

Reviewer Point P 1.1 —From a machine learning point of view, the tasks that can be performed by training RNNs using the predictive alignment method can all be implemented by the FORCE method and its variants, often with better performance; some of these papers are cited in the manuscript. From a biological perspective, the methods developed in this study lack the biological plausibility of training RNNs with spiking neurons. Regarding this latter point, the e-prop method (Bellec et al., Nature Communications, 2020), which is also local and leverages error signals, offers more biological plausibility. As the field stands now, a powerful new method that can effectively train recurrent spiking neural networks would represent significant progress.

Reply: Regarding the first point, while it is true that the tasks used in our study can all be performed by FORCE learning in most cases, we would like to emphasize that we have shown the proposed model performs even better than FORCE in the presence of noise (Supplementary Figure 2, shown again below). This happens because FORCE aims to make its output match the target signal as quickly as possible, which can be disrupted by strong noise. In contrast, our model learns more slowly, making the learning process less affected by noise. We have explained these results in ll. 227-231 in the revised manuscript.

Figure R1. Robustness of predictive alignment against noise. (A) Networks with different strengths of noise were trained with the FORCE (red) and the predictive alignment (green) with the patterned target signal. Error bars stand for s.d.s over 20 independent simulations. (B) Example readout activities during the late phase of training are shown. Colors are the same as in A. The gray traces represent the target signal.

To address the second point, we asked whether our learning rule can be implemented in a spiking recurrent network. We implemented a recurrent network composed of 1,000 leaky integrate-and-fire (LIF) neurons, coupled with a single linear readout unit as the output. We ran additional simulations and found that our rule can be applied in a network of spiking neurons (Fig. R2 shown below). We have included the new results as a new Supplementary Figure 8 in the revised manuscript. The details of simulation results and implementations are shown in II. 494-500 and II.756-783 in the revised manuscript, respectively.

Figure R2. The spiking recurrent network was trained to learn periodic target signal. (A) The blue trace represents the target signal, and the green line represents the output. (B) Activities of spiking network neurons before training are shown. (C) Activities of trained spiking network neurons are shown. The modified recurrent connections generate the sequential network activity.

Reviewer Point P 1.2 — It is unclear why the learning method works. In other words, what changes occur during the learning process that enable the RNNs to perform the intended computations? Are these changes related to the reorganization of the underlying chaotic attractor into a rich combination of fixed points or continuous attractors of the corresponding dynamical systems? I would like to see a deeper treatment of this question to open the “black box” of this process. The learning performance is best at the “edge of chaos,” which is a very interesting observation, but why do such critical dynamics facilitate learning?

Reply: The reviewer’s intuition is correct. Our rule suppresses the chaotic spontaneous activity and reorganize it to multiple attractors by modifying the recurrent connections. To address this, we adopted the approach of Sussillo and Barak (2013), where three output units were specifically trained to read out the corresponding fixed points of a network dynamics. The state of each output is determined by transient pulses from the corresponding input units. These pulses affect the corresponding output unit, causing it to switch or maintain a value of either +1 or -1. Once the output value is set, it remains fixed until the arrival of the next pulse from the input unit. We found that the trained network is capable of producing outputs that show transition between states in response to input (Figure R3A).

To observe how the trained network switches between memory states, we perturbed the state of the trained network with input pulses to observe transitions between two fixed points over six trials. In these trials, the strength of the input was gradually increased. When the input was weak, we found that the network activity briefly deviated from its fixed point and returned to the original fixed point when the input was removed. In striking contrast, when the input was strong enough, the network switched to a different fixed point, suggesting the existence of saddle points between the two stable fixed points, demonstrating its mechanistic role in the corresponding transition (Figure R3B). We have included these additional simulation results as Supplementary Figure 5 and explained the

results in II.325-358 and implementation details in II.747-754 in the revised manuscript.

Furthermore, we have included an additional sentence to explain why the edge of chaos is important in the proposed model in II. 310-312: “The model performs best at the edge of chaos because it supports richness of network dynamics, which in turn provides rich basis functions from which the output units can readout the appropriate dynamics.”.

Figure R3. Learning fixed point attractors. (A) Example three inputs (black) and outputs (blue, orange, and green) are shown. (B) Low dimensional network dynamics are shown. Network dynamics perturbed with the varying strength of inputs shows that a saddle point mediates the transitions between attractors.

Reviewer Point P 1.3 — What is the biological relevance of assuming the recurrent connectivity is a summation of M and G?

Reply: The reviewer raised an important point. Our plasticity rule assumes two types of recurrent connections (i.e., M and G), one is plastic and the other is static. While the biological relevance of this is still an open question, it is known experimentally that the degree of synaptic plasticity varies across different compartments of dendrites (Gordon et al., J Neurosci. 2006). Based on this experimental evidence, we can speculate that the plastic connections are synapses that project onto proximal basal dendrites and the static connections

project onto distal basal dendrites. We have included this point in II.631-637 in the revised manuscript.

Reviewer Point P 1.4 — What are the effects of certain hyperparameters, such as learning rates, on learning performance? Without thorough exploration of this aspect, it is difficult to assess whether the learning method is robust and effective.

Reply: We have investigated the robustness of the model with respect to its hyperparameters. First, we examined the effect of the learning rate on performance. We found that as the learning rate increased, so did the output error; however, if the learning rate was sufficiently small, the error remained small even when the learning rate was increased tenfold (Figure R4A). Next, we examined the initial strength of the plastic recurrent connections, where the connections were fully connected. We found that as long as the initial connection strength was not extremely large (e.g., when it was larger than that of a strong connection G), the error remained sufficiently small (Figure R4B). Finally, we examined against the connection probability of M and found that as long as the connections were not extremely sparse, the error remained sufficiently small (Figure R4C). These results suggest that the learning performance was not susceptible to certain hyperparameter choices. We have included these new results as a new Supplementary Figure 3 and explained in II. 233-237 in the revised manuscript.

Figure R4. Robustness to hyperparameter choices. (A) The networks were trained with different levels of learning rates. (B) Same as A, but trained with different degrees of the initial strength of the plastic recurrent weights. (C) Same as A, but trained with different

degrees of connection probabilities of the plastic recurrent weights. Error bars represent SEs across five independent simulations.

Reviewer #2

Reviewer Point P 2.1 —My primary concern relates to the apparent absence of noise in the RNN. As far as I could tell by looking at the Methods and sample code (perhaps I'm wrong), in contrast to most RNN models, there was no noise term during training or testing. When proposing a biologically plausible learning rule that relies on feedback from a chaotic RNN the absence of noise is potentially a serious issue, because training may not work at all when noise is present as the feedback signal is a moving target. It is thus necessary to include noise and parametrically vary noise to understand the influence of noise in predictive alignment.

Reply: We apologize to the reviewer for any confusion caused by our manuscript. We had already included external noise in the network during learning. In the Supplementary Figure 2 (shown again below), we have shown that the proposed model performs even better than FORCE in the presence of noise. These results indicate that our model is robust to external noise in the network. We have explained these results in ll. 227-231 in the revised manuscript.

In the revised manuscript, we clarified this point by including noise term explicitly in the Equation 7, and mentioned “We will consider network dynamics with external drives and noise.” in ll. 84-85 in the Results section to avoid confusion.

Figure R5. Robustness of predictive alignment against noise. (A) Networks with different strengths of noise were trained with the FORCE (red) and the predictive alignment (green) with the patterned target signal. Error bars stand for s.d.s over 20 independent simulations. (B)

Example readout activities during the late phase of training are shown. Colors are the same as in A. The gray traces represent the target signal.

Reviewer Point P 2.2 —The use of the Measure-Wait-Go task is interesting, and an important demonstration of the power of the learning rule. However, the time scale used was less than an order of magnitude longer than that of τ . Does the approach work for more biologically relevant time scales such as those used in the Jazayeri and Shadlen paper: hundreds of milliseconds? Also, this task is normally referred to as the ready-set-go task (Jazayeri and Shadlen, 2010). It would also be helpful to show a sample of the RNN activity during the task rather than just the PC's, because linear PCs can emerge from RNNs without much linear activity.

Reply: We increased the delay period to be remembered in the task to more than 100 ms. We found that although the delay period was more than 10 times the time scale of the neuron's dynamics, the model still showed the same behavior as in the previous setting. In the revised manuscript, previous Figure 5 has been replaced by the new simulation results. In a supplementary Figure 7, we have shown ten samples of the RNN activity in the revised manuscript (shown as Fig. R7 below). Furthermore, in the revised manuscript, we have changed the term "Measure-Wait-Go task" to "Ready-Set-Go task".

Learning over much longer timescales (i.e., hundreds of milliseconds) can in principle be achieved by extending network timescales through larger networks, adaptation, and short-term plasticity. We want to leave this as a future work.

Figure R6. Learning delay-matching task. (A) In each trial, two input units (blue and orange) send pulses to the network with a random delay T_{delay} between pulses. The target network output (green) should be a pulse delayed by T_{delay} relative to the second input pulse. (B) Networks were trained on a set of samples (colored squares) and tested for generalization to novel inputs, including interpolation within the training range (shaded region) as well as extrapolation beyond the training range. (C) The network after training failed to extrapolate beyond the training range. (D) Principal component analysis (PCA) of the trained network revealed that as the delay T_{delay} increased, the network states corresponding to the output peak shifted linearly along a manifold in state space.

Figure R7. Neural activities for performing Ready-Set-Go task. (A) In each trial, two input units (blue and orange) send pulses to the network with a random delay T_{delay} between pulses. The target network output (green) should be a pulse delayed by T_{delay} relative to the second input pulse. (B) Networks were trained on a set of samples (colored squares) and tested for generalization to novel inputs, including interpolation within the training range (shaded region) as well as extrapolation beyond the training range. (C) The network after training failed to extrapolate beyond the training range. (D) Principal component analysis (PCA) of the trained network revealed that as the delay T_{delay} increased, the network states corresponding to the output peak shifted linearly along a manifold in state space.

Reviewer Point P 2.3 —Replay task. It was not clear what time = 0 was for the movie generation? Does the produced movie run in a limit cycle?

Reply: We thank the reviewer for pointing this out. At time zero, we set the network activity to the predefined initial states. The same initial state was used in both the learning and testing phases. We have included these explanations in ll. 515-518 in the revised manuscript. While we reset the initial state over trials in the results shown in Figure 6, in principle the movie can run in a limit cycle and restart when the movie ends.

Reviewer Point P 2.4 —Pg. 2. To support the statement “Cortical circuits often exhibit chaotic spontaneous activity” a number of papers are cited, but they all seem to be models, not experimental papers (and I don’t think Rajan et al, 2016 even address chaos). RNNs are not cortical circuits so these references do not seem appropriate. London, ..., Latham, 2010 provides some evidence for chaos in cortical circuits.

Reply: The reviewer is correct, we cited only computational papers in our previous manuscript. We have included the suggested reference and changed the sentence as: “Cortical circuits often exhibit chaotic spontaneous activity (London et al., 2010), and RNNs can generate such dynamics through feedback loops (van Vreeswijk and Sompolinsky, 1996; Amit and Brunel, 1997; Brunel, 2000; Toyozumi and Abbott, 2011; Rajan et al., 2016).”, in ll. 42-45 in the revised manuscript. Here, we have kept the Rajan et al. paper since they used RNNs showing chaotic behavior before training.

Reviewer Points P 2.5 —The presentation and discussion omitted many relevant studies that laid the foundation for the current work.

Reply: We hope we have included the appropriate references in the revised manuscript. If there are any missing, could the reviewer let us know?

Reviewer Points P 2.6 —Jaeger and Haas 2004, essentially published what later came to known as FORCE in their “online” learning rule.

Reply: We have included the reference in the revised manuscript.

Reviewer Points P 2.7—The paper has commonalities with Laje and Buonomano, which also took the approach of taming chaos by changing the recurrent weights and using high dimensional structure of a local stable trajectories as a reservoir. That work was not cited in the text. And Buonomano and Merzenich1995, was the first description of what later came to be called a reservoir network.

Reply: We have included the references in the revised manuscript.

Reviewer Points P 2.8—Supplemental Figure 4. A schematic of the network would be helpful.

Reply: We have included new schematics to help readers. We thank the reviewer for pointing this out.

Reviewer Points P 2.9—Pg. 14 “while failed AT extrapolation” ; Pg. 15 “Here, we train recurrent neural” ... typo.

Reply: We have corrected the typo. We thank the reviewer for their careful review.

Reviewer Points P 2.10—It is best to refer to the units and weights as such, rather than neurons and synapses.

Reply: We agree with the reviewer to change our terminology. We have changed the terminology in the revised manuscript.

We look forward to hearing from you soon. Thank you very much for your kind consideration in advance.

Sincerely yours,

Toshitake Asabuki
RIKEN Center for Brain Science, Japan

toshitake.asabuki@riken.jp

Prof. Claudia Clopath

Department of Bioengineering, Imperial College London, London, UK.

c.clopath@imperial.ac.uk; +44 (0)20 7594 1435

Response to the reviewers

Toshitake Asabuki and Claudia Clopath

We thank the reviewers for their thorough and critical evaluation of our work. We read the reviewers' comments carefully and we **think we have addressed reviewers' concerns fully**. Below, we explain how we addressed each of their concerns point-by-point. Thank you very much in advance for your time.

Reviewer #1

Reviewer Point P 1.1 —I appreciate the authors' detailed responses and the corresponding revisions. However, my concerns regarding the training of RNNs with spiking neurons and the changes that occur during the learning process to enable these networks to perform the intended computations remain unclear. Specifically:

(1) For the proposed learning rule to be effectively evaluated in a spiking neural network, it is essential to compare its performance with existing methods for training SNNs. Additionally, applying the proposed method to balanced spiking neural networks with both excitatory and inhibitory neurons would be more appropriate for two reasons: (i) it aligns better with biological plausibility, and (ii) balanced networks are known to exhibit chaotic dynamics, which are more consistent with the problem addressed in this study.

Reply: That's a great suggestion. We have compared the performance with two types of state-of-the-art spiking recurrent network for supervised learning: FORCE and e-prop. We have included the new results as Supplementary Fig. 10C and mentioned in ll.524-525 in the revised manuscript.

Furthermore, we showed that our proposed rule enables learning even in a recurrent spiking neural network composed of two populations (i.e., excitatory and inhibitory). We have included the new result as Supplementary Figure 11 and mentioned in ll.525-528 in the revised manuscript.

Reviewer Point P 1.2 —(2) While the revision presents some results on learned fixed points, it remains unclear how and why the underlying chaotic dynamics are reorganized during the training process.

Reply: The reviewer raised an important point. While we have shown that aligning the predictive recurrent and chaotic dynamics is important for stable learning in Figure 3, we did not explain how it changes the recurrent dynamics. To see this further and understand how the alignment enables stable learning, we monitored the network's Lyapunov exponent during training. We found that in the aligned case, the network's Lyapunov exponent was shifted further toward the negative side compared to the control case, indicating more effective suppression of chaos. We have included the new simulation result as Supplementary Figure 4 and explained in II. 270-272 in the revised manuscript.

Reviewer Point P 1.3 —(3) The added explanation for why the model performs best at the edge of chaos is largely qualitative and descriptive (see Line 310). In machine learning, it is well established that artificial neural networks achieve optimal training performance near the edge of chaos [Schoenholz, et al., Deep information propagation, ICLR 2017], supported by quantitative analysis. Similarly, I believe a quantitative investigation is necessary to strengthen this argument.

Reply: To investigate why learning performance is maximized at the edge of chaos, we analyzed the eigenvalue distribution of the correlation matrix of the network activity. Optimal learning is thought to require a balance between representational diversity and dimensionality. High diversity allows the network to encode rich information, while low dimensionality ensures compact representations. We quantified the diversity of neural representations using the entropy of the eigenvalues of the correlation matrix (H_λ), and measured the balance between information spread and localization using the square root of the participation ratio (\sqrt{PR}). We computed the ratio of these two measures,

$$\text{Efficiency} = \frac{H_\lambda}{\sqrt{PR}}$$

and found that this balance index was maximized specifically at the edge of chaos. In the subcritical regime, the participation ratio was low, indicating that activity was concentrated in a low-dimensional subspace, reducing the network's ability to learn diverse representations. In contrast, in the chaotic regime, the participation ratio was high, but the eigenvalue entropy decreased, suggesting

excessive dispersion of information, leading to loss of meaningful structure. At the edge of chaos, both measures were optimally balanced, enabling the network to achieve the highest learning performance. These findings suggest that learning is most efficient when the network exhibits rich yet structured dynamics, balancing representational diversity and stability. We have included the new simulation results as Supplementary Figure 6 in the revised manuscript and explained in ll. 327-347 in the revised manuscript.

Reviewer #2 (Remarks to the Author):

The authors have adequately addressed my concerns, and the inclusion of the spiking novel further improves the paper?

"matrixes" should be "matrices"

Reply: We thank the reviewer for the careful review. We have fixed the typo.

We look forward to hearing from you soon. Thank you very much for your kind consideration in advance.

Sincerely yours,

Toshitake Asabuki
RIKEN Center for Brain Science, Japan
toshitake.asabuki@riken.jp

Prof. Claudia Clopath
Department of Bioengineering, Imperial College London, London, UK.
c.clopath@imperial.ac.uk; +44 (0)20 7594 1435

Response to the reviewers

Toshitake Asabuki and Claudia Clopath

Reviewer #1

Reviewer Point P 1.1 —In the revised manuscript, the authors have included comparisons between the proposed learning rule and two other supervised learning methods, and have added some quantitative explanations concerning why learning performance is maximised at the edge of chaos. My main concerns have now been addressed in the revisions.

Reply: We thank the reviewer for their fruitful feedback.

We look forward to hearing from you soon. Thank you very much for your kind consideration in advance.

Sincerely yours,

Toshitake Asabuki
RIKEN Center for Brain Science, Japan
toshitake.asabuki@riken.jp

Prof. Claudia Clopath
Department of Bioengineering, Imperial College London, London, UK.
c.clopath@imperial.ac.uk; +44 (0)20 7594 1435